# Perfusion Techniques in Kidney Allograft Preservation to Reduce Ischemic Reperfusion Injury: A Systematic Review and Meta-Analysis

**DOI:** 10.3390/antiox13060642

**Published:** 2024-05-24

**Authors:** Bima J. Hasjim, Jes M. Sanders, Michael Alexander, Robert R. Redfield, Hirohito Ichii

**Affiliations:** 1Department of Surgery, Division of Hepatobiliary and Pancreas Surgery and Islet Cell Transplantation, University of California–Irvine, Orange, CA 92868, USA; bhasjim@hs.uci.edu (B.J.H.); michaela@hs.uci.edu (M.A.);; 2Department of Surgery, Division of Transplantation, Northwestern Memorial Hospital, Chicago, IL 60611, USA; jes.sanders@nm.org

**Keywords:** machine perfusion, normothermic regional perfusion, ischemic reperfusion injury, antioxidants, organ preservation, static cold storage, in situ cold preservation

## Abstract

The limited supply and rising demand for kidney transplantation has led to the use of allografts more susceptible to ischemic reperfusion injury (IRI) and oxidative stress to expand the donor pool. Organ preservation and procurement techniques, such as machine perfusion (MP) and normothermic regional perfusion (NRP), have been developed to preserve allograft function, though their long-term outcomes have been more challenging to investigate. We performed a systematic review and meta-analysis to examine the benefits of MP and NRP compared to traditional preservation techniques. PubMed (MEDLINE), Embase, Cochrane, and Scopus databases were queried, and of 13,794 articles identified, 54 manuscripts were included (*n* = 41 MP; *n* = 13 NRP). MP decreased the rates of 12-month graft failure (OR 0.67; 95%CI 0.55, 0.80) and other perioperative outcomes such as delayed graft function (OR 0.65; 95%CI 0.54, 0.79), primary nonfunction (OR 0.63; 95%CI 0.44, 0.90), and hospital length of stay (15.5 days vs. 18.4 days) compared to static cold storage. NRP reduced the rates of acute rejection (OR 0.48; 95%CI 0.35, 0.67) compared to in situ perfusion. Overall, MP and NRP are effective techniques to mitigate IRI and play an important role in safely expanding the donor pool to satisfy the increasing demands of kidney transplantation.

## 1. Introduction

Chronic kidney disease (CKD) is estimated to affect more than 840 million individuals worldwide [1] with a global prevalence of 13.4% [2]. Emerging as one of the leading causes of death, CKD’s global mortality rate has increased by 41.5% over the past three decades [3,4]. Currently, kidney transplantation is the only definitive treatment for CKD once it progresses to its end stages but is limited by the availability of organs. Among patients waitlisted in 2016–2018, 19.4% were removed from the waitlist without a transplant, 33.0% were still waiting, and 6.2% died without a transplant after 3 years of follow-up [5].

### 1.1. Use of Allografts Susceptible to Oxidative Stress and Ischemic Reperfusion Injury (IRI)

The current demand for kidney grafts for transplantation far outweighs the supply. Thus, it is imperative to innovate new ways to increase the deceased donor pool. Grafts from less ideal donors such as extended criteria donors (ECDs) and donation after circulatory death (DCD) are viable options to help address the demand. However, these higher risk grafts are more vulnerable to greater degrees of ischemic reperfusion injury (IRI) and oxidative stress due to differences in metabolic demand. During the process of organ procurement, preservation, and transplantation, allografts accumulate a series of physiologic insults that are paradoxically exacerbated at the time of organ reperfusion (Figure 1). The first insult occurs during organ procurement when blood flow to the organ is discontinued, when cells transition from aerobic to anaerobic metabolism [6].The altered physiology under prolonged ischemic conditions [7] leads to oxidative stress, decreased adenosine triphosphate (ATP) production, and accumulation of intracellular lactate. Uncompensated acidosis ensues with a subsequent imbalance of electrolytes and cellular swelling. Upon restoration of blood flow at the completion of transplantation, although oxygen availability and acidosis are normalized, a second insult occurs. Intracellular proteinases, permeability of the mitochondrial membrane, and generation of reactive oxygen species (ROS) are activated leading to further oxidative stress and cell injury [8]. Paradoxically, the sum-total effect of these processes leads to cellular necrosis and programmed cell death from activation of apoptotic, ferroptotic, and autophagy pathways. Therefore, a key aspect of expanding the donor pool relies on innovative organ preservation techniques to reduce IRI.

### 1.2. Organ Preservation Techniques: Innovating the Gold Standard

Static cold storage (SCS) is the oldest technique of organ preservation and has been the standard of care since the 1970s. SCS involves submerging the kidney allograft in preservation solution that typically includes colloids, antioxidants, and molecular precursors of ATP at the time of procurement [9]. These allografts are stored in an insulated container of preservation solution and melting ice to create a hypothermic environment until the time of transplantation. More recently, dynamic preservation techniques have emerged as an improved alternative to mitigating IRI, especially among higher risk donors. Machine perfusion (MP) [10] and normothermic regional perfusion (NRP) [11] are two different modalities of dynamic preservation that have gained the interest of clinical trial design. MP involves the perfusion of (1) cooled preservation solution through the organ, as in hypothermic machine perfusion (HMP), or (2) oxygenated perfusate/blood warmed to body temperature (35–38 °C), as in normothermic machine perfusion (NMP). HMP has been shown to reduce cellular metabolism by 5–10%, and its hemodynamic stimulus maintains the cortical microcirculation during the preservation period [12,13]. Alternatively, NMP aims to mimic the physiologic environment during ex situ preservation and has the ability to rehabilitate allografts and evaluate graft function prior to transplantation [14,15]. Lastly, NRP is deployed during DCD donor retrieval through an extracorporeal membrane oxygenation (ECMO) device that restores circulation of oxygenated blood after cardiac arrest and before organ procurement. NRP is often compared with in situ cold preservation (ISP) after rapid recovery, which is a single perfusion of cooled preservation solution. NRP has the added benefit of reducing warm ischemia time and providing a controlled organ-retrieval process that ultimately protects against hypoxic injury.

Assessing these different organ preservation techniques to mitigate organ dysfunction as more borderline grafts are considered has motivated several clinical trials in the past decade. While delayed graft function and other perioperative and short-term outcomes have been the primary focus of prior reviews and meta-analyses [10,13], it is also imperative to evaluate long-term outcomes and biomarker endpoints. Noninvasive biomarkers, such as serum creatinine or estimated glomerular filtration rate (eGFR), can help anticipate the clinical trajectory of a graft prior to conventional clinical endpoints. To this end, we conducted a comprehensive, systematic review and meta-analysis to evaluate the outcomes of contemporary perfusion techniques.

## 2. Materials and Methods

### 2.1. Search Strategy 

The following review follows the Preferred Reporting Items for Systematic Reviews and Meta-analyses Protocols (PRISMA-P) [16] and was registered with the PROSPERO database of systematic review protocols (PROSPERO ID: CRD42024506080). PubMed (MEDLINE), Embase, Cochrane, and Scopus databases were queried with the help of a biomedical information specialist. The following search terms were included: “cold storage”, “machine perfusion”, “normothermic regional perfusion”, “chronic kidney disease”, “kidney transplantation”, “delayed graft function”, “primary nonfunction”, “ischemia reperfusion injury”, “graft rejection”, and “mortality”. The complete search strategy is detailed in the Appendix A.

### 2.2. Study Selection and Eligibility Criteria

Two authors (BJH and JMS) independently screened each study’s title, abstract, and full text using an online standardized tool. Titles/abstracts were selected for review based on relevance. Discrepancies were reconciled through discussion and/or referral to a third reviewer (HI) if needed. Manual inclusion of relevant references was performed after the review of each study’s reference list to ensure no relevant articles were missed.

Articles from the inception of each database until 6 February 2024 were included if data could be extracted, they were published in English, and full texts were available. Studies without quantitative data, relevant comparison groups, review papers, and those not published in a peer-reviewed journal (e.g., abstracts, preprint letters, editorials, dissertations, book sections, film or broadcasts, opinion articles, and guidelines) or those involving case reports, case series, or basic science studies were excluded.

### 2.3. Study Population and Outcomes of Interest

A data extraction form was used to collect study details, donor and recipient characteristics, and outcomes from each study. Study details included authors, publication year, multicenter involvement, kidney pair involvement, sample size of cohorts, and study design. Organ donor characteristics included donor type, mean cold ischemia time (CIT), warm ischemia time (WIT), and age. Recipient characteristics and outcomes of interest included age, hospital length of stay, delayed graft function (DGF), primary nonfunction (PNF), and acute rejection. Long-term outcomes include serum creatinine (mg/dL; at 3, 6, and 12 months), estimated glomerular filtration rate (eGFR; ml/min; at 3, 6, and 12 months), graft failure (at 12 months and 3 years), and patient mortality (at 12 months and 36 years). The DCD type, either from a controlled (cDCD) or uncontrolled (uDCD) setting, was reported if available. Grafts from ECD were defined as those from a donor with any of the following criteria: (1) any brain-dead donor (a) aged > 60 years or (b) aged 50–60 years with at least two of the following: history of hypertension, terminal serum creatinine level ≥ 1.5 mg/dL, death resulting from a cerebrovascular accident; (2) prolonged (>20 min) donor circulatory arrest; (3) cold ischemia time > 24 h [17].

The long-term outcomes of MP and NRP were compared to SCS and ISP, respectively, among kidney transplant recipients. The primary outcome was 12-month graft survival and post-transplant biomarker surrogates for renal function (e.g., eGFR, serum creatinine). Secondary outcomes were DGF, PNF, acute rejection, and mortality. DGF was defined as the need for dialysis within 14 days after kidney transplantation, whereas PNF was defined as the lack of any graft function after kidney transplantation. There were not enough data to analyze the difference in 3-year mortality among HMP studies. Data on 3- and 6-month serum creatinine and 6-month eGFR were also omitted from analysis among NRP studies due to a lack of extractable data.

### 2.4. Statistical Analysis

Pooled categorical data were presented as percentages, while continuous variables were presented with means, each with their corresponding 95% confidence interval (95%CI), respectively. Missing means and standard deviations were estimated by the sample size, median, interquartile range, and range as previously published [18,19]. Studies with reported means and medians but without other components for analysis were not included in the final analysis but were maintained in the dataset and forest plots for completeness. Only studies where MP and NRP were compared to SCS and ISP, respectively, were included. A common effect model (or fixed-effect method) was used for the analysis of more statistically homogenous data (I^2^ < 50%), while a random effect model was used to analyze and present data for more heterogenous data (I^2^ > 50%). Both common effect and random effect models were generated for each outcome as sensitivity analyses to ensure there were no significant differences. The Mantel–Haenszel approach was used to analyze categorical data, while a standard inverse variance approach was used for continuous data. Meta-analyses of continuous and categorical variables were represented by their mean difference (MD) or odds ratio (OR) with their respective 95%CI. An alpha level of <0.05 was considered to be statistically significant. Data processing and analysis were performed using R studio (version 4.2.2; Vienna, Austria).

## 3. Results

### 3.1. Overview of the Study Selection Process

A total of 13,794 articles were identified through database retrieval. After removing duplicates (N = 4569 articles) and screening titles/abstracts (N = 9225 articles), 90 articles were included for full-text review. Of these, 36 did not meet the selection criteria and were excluded. In total, there were 55 manuscripts included (N = 42 MP vs. SCS articles [9,15,17,20,21,22,23,24,25,26,27,28,29,30,31,32,33,34,35,36,37,38,39,40,41,42,43,44,45,46,47,48,49,50,51,52,53,54,55,56,57]; N = 13 NRP vs. ISP articles [58,59,60,61,62,63,64,65,66,67,68,69,70]) (Figure 2).

### 3.2. Machine Perfusion (MP) vs. Static Cold Storage (SCS) 

A total of 42 studies [9,15,17,20,21,22,23,24,25,26,27,28,29,30,31,32,33,34,35,36,37,38,39,40,41,42,43,44,45,46,47,48,49,50,51,52,53,54,55,56,57,71] reported the outcomes related to MP. A total of 39 studies investigated the effects of HMP [9,17,20,21,22,23,24,25,26,27,28,29,30,31,32,33,34,35,36,37,38,39,40,41,42,43,44,45,46,47,48,49,50,51,52,53,54,55,56], while 2 studies focused on NMP [15,72]. Among MP studies, 10 (24.4%) were retrospective studies [23,29,37,38,39,42,43,49,50,54], 17 (41.5%) were prospective studies [9,17,21,22,24,25,26,27,28,35,40,41,46,48,55,57], and 14 (34.2%) were randomized control trials [15,20,30,31,32,33,34,36,44,47,51,52,53,56]. There were 22 (53.7%) studies that involved kidney pairs [17,20,21,22,24,28,30,31,32,33,34,36,41,44,45,46,47,48,52,57]. Studies were conducted across the world; 25 (61.0%) were single-center efforts [9,17,22,24,25,26,27,28,29,30,35,38,39,40,41,42,43,45,46,48,49,51,54,55,57], while 16 (39.0%) involved multiple centers [15,20,21,23,31,32,33,34,36,37,44,47,50,52,53,56]. These studies encompassed 57,756 patients who received grafts maintained by MP compared to 82,956 patients with SCS. Donor (50.0 years vs. 49.8 years, *p* = 0.702) and recipient (50.6 years vs. 50.8 years, *p* = 0.776) ages for MP and SCS were comparable. MP grafts had longer CIT (15.4 h vs. 13.4 h, *p* = 0.005), WIT (33.9 min vs. 30.8 min, *p* = 0.016), and higher terminal donor serum creatinine (1.16 mg/dL vs. 1.09 mg/dL, *p* = 0.030) compared to SCS.

DGF was the most commonly investigated outcome (90.2%), followed by 12-month graft failure (56.1%) and acute rejection (53.7%). Grafts maintained by MP had lower rates of DGF (27.7% vs. 36.4%, *p* = 0.001) and 12-month graft failure (5.3% vs. 8.3%, *p* < 0.001). Among those who suffered from DGF, there were no differences in the duration of DGF or mean number of dialysis procedures (*p* > 0.05). MP also had lower rates of PNF (2.8% vs. 4.4%, *p* = 0.011) and shorter mean lengths of hospital stay (15.5 days vs. 18.4 days, *p* = 0.009) compared to SCS. There were no differences in rates of acute rejection, 3-year graft failure, and 12-month or 3-year mortality (*p* > 0.05) (Table 1).

MP had lower odds of 12-month graft failure (OR 0.66; 95%CI 0.55, 0.79) (Figure 3), DGF (OR 0.68; 95%CI 0.54, 0.86) (Figure 4), and PNF (OR 0.63; 95%CI 0.44, 0.90) (Figure 5). MP had lower mean lengths of hospital stay (MD −2.63, 95%CI −4.61, −0.66) compared to SCS (Figure 6).

There were fewer studies that assessed graft function through biomarkers such as serum creatinine (29.3%) and eGFR (24.4%). MP and SCS grafts had comparable serum creatinine levels at 3 months (1.74 mg/dL vs. 1.90 mg/dL, *p* = 0.063), 6 months (1.67 mg/dL vs. 1.70 mg/dL, *p* = 0.226), and 12 months (1.58 mg/dL vs. 1.66 mg/dL, *p* = 0.278) post-transplant. Likewise, eGFR at all post-transplant time points were similar between MP and SCS (*p* > 0.05) (Table 1) (Figure 7).

### 3.3. Normothermic Regional Perfusion (NRP) vs. In Situ Cold Preservation (ISP)

There were 13 articles that compared NRP vs. ISP [15,57,59,63,67,68]. Among NRP studies, there were 6 (46.2%) retrospective studies [59,60,61,67,68,70] and 7 (53.8%) prospective studies [58,62,63,64,65,66,69], but no (0%) randomized control trials. Nine (69.2%) studies were from single center experiences [58,59,60,61,62,64,66,67,69], while four (30.8%) were multicenter efforts [63,65,68,70]. There were 2025 and 3229 patients receiving grafts treated with NRP and ISP, respectively. The NRP and ISP cohorts had similar mean donor age (48.1 years vs. 47.8 years, *p* = 0.903), though NRP recipients were older (50.1 years vs. 49.4 years, *p* = 0.042) compared to ISP. The NRP cohort had shorter mean CIT (12.7 h vs. 17.2 h, *p* = 0.001) and WIT (22.5 min vs. 24.4 min, *p* < 0.001). There were no differences in terminal donor serum creatinine between NRP and ISP (1.34 mg/dL vs. 1.29 mg/dL, *p* = 0.639) (Table 2).

DGF was the most commonly investigated outcome (85.7%), followed by PNF (76.9%) and 12-month graft failure (76.9%). A meta-analysis of the results from these studies showed that there were no differences in rates of DGF, PNF, length of hospital stay, or 12-month or 3-year graft failure or mortality (*p* > 0.05). NRP had a lower incidence (6.4% vs. 10.0%, *p* < 0.001) (Table 2) and odds (OR 0.48; 95%CI 0.35, 0.67) (Figure 8) of acute rejection compared to ISP. There were no differences in serum creatinine or eGFR levels at 3- or 12-month post-transplant follow-up (Table 2) (Figure 9).

## 4. Discussion

### 4.1. Summary of Results, Implications

Organ preservation techniques must keep pace with the expanding donor pool as the transplant community strives to satisfy the growing, worldwide demand of kidney transplantation. Thus, it is important to elucidate its benefits, opportunities, and limitations to optimize these perfusion devices. In our meta-analysis, the inclusion of a wide range of study types and a large cohort of patients allows our analysis to detect rare outcomes that other meta-analyses may not have been able to detect [10,13,73]. This is particularly important when investigating long-term outcomes and, to our knowledge, this is the first meta-analysis to investigate the long-term clinical and serological outcomes of various organ preservation techniques in kidney transplants. Overall, MP and NRP were associated with superior and non-inferior graft outcomes compared to traditional preservation techniques, respectively. Not only does MP reduce the incidence of poor short-term outcomes (e.g., DGF, length of hospital stay, acute rejection, PNF) compared to SCS, it also reduces the incidence of 12-month graft failure. In contrast, although NRP- and ISP-treated grafts were similar in most clinical outcomes, NRP was associated with a reduction in rates of acute rejection, leaving reason to believe that there are further benefits to NRP barring future research.

Though it has been slow to be implemented in the US, MP has emerged as the superior mode of organ preservation in kidney transplantation and has been adopted as the standard of care in many countries within the last decade. Similar to prior meta-analyses [10,13,73], our results show that MP is associated with lower odds of DGF and PNF by 32% and 37%, respectively. These findings robustly hold true despite donors from the MP cohort having higher risk factors that contribute to poor outcomes [74]: grafts maintained by MP more often came from donors with higher mean terminal serum creatinine (1.16 mg/dL vs. 1.09 mg/dL, *p* = 0.014) and longer mean CIT (15.2 h vs. 13.3 h, *p* < 0.001) and WIT (33.9 min vs. 30.8 min, *p* = 0.024). In a meta-analysis of only randomized controlled trials (RCTs), the number of MP-treated grafts required to prevent one episode of DGF among DCD and donation after brain death (DBD) grafts was 7.26 and 13.60 patients, respectively [73]. Our study also found that grafts maintained by MP had lower rates of graft failure at 12 months post-transplant (OR 0.66; 95%CI 0.55, 0.79). These data confirm the benefits of MP in mitigating the short-term effects of IRI, but also in sustaining these benefits through long-term follow-up. The occurrence of DGF, especially prolonged periods of DGF, can increase the risk of graft failure and overall mortality by 70% and 80%, respectively [74,75]. Furthermore, another meta-analysis found that patients with DGF had a 41% increased risk of graft loss compared to those without DGF [76]. Overall, MP can effectively mitigate the destructive effects of IRI as evidenced by its superior short- and long-term graft function, even if subjected to high risk donors.

In addition to traditional clinical outcomes, it is also important to consider the utilization of resources that result from these perfusion techniques. In our meta-analysis, we found that MP grafts had lower mean hospital lengths of stay compared to SCS (15.5 days vs. 18.4 days, *p* = 0.009). Length of stay has been correlated with higher rates of DGF and PNF due to the increased need for renal replacement therapy and further assessment of graft function [77]. Despite the costly investment of MP devices in the perioperative period ($1182 MP vs. 234 SCS), estimates of costs at 12-month follow-up showed that MP was ultimately $3627 cheaper than SCS per transplant ($8668 vs. 11,394) [78]. Gomez et al. estimated as much as $3369 in savings for each DGF or PNF avoided by MP [79]. The main contributors to the disparity in costs between MP and SCS were from post-transplant dialysis requirements ($4390 MP vs. 7581 SCS) and hospital readmissions ($2892 MP vs. 3174 SCS) [78]. The cost–benefit of MP is further amplified among ECD compared to standard criteria donors (SCDs). In ECD cases, the yearly cost–benefit of MP is 35.7% greater than the cost-savings of SCD cases at 12-month post-transplant follow-up [80]. Currently, data are lacking for a formal meta-analysis on the cost–benefit of MP, and future research should consider it as a primary endpoint as the use of perfusion devices becomes more prevalent.

In contrast to the clinical benefits of MP, the only observed difference in outcomes between NRP and ISP was the rate of acute rejection. In our meta-analysis, NRP reduced the odds of acute rejection by 52%. While there were no significant differences in DGF and long-term graft loss, it should not be interpreted that NRP is ineffective in mitigating IRI. IRI is known to promote the activation of antigen-presenting cells and increase production of alloantibodies which leads to acute rejection [81,82]. Therefore, a reduction in rates of acute rejection may act as a partial mediator between DGF and subsequent graft loss [75,83]. The lack of significant differences in DGF, PNF, and graft survival should also be interpreted with caution, as there were high degrees of heterogeneity in the study population. Also, NRP is a relatively newer technology with maturing literature. The limited scope of studies in NRP vs. ISP, compared to the collection of HMP vs. SCS studies, has an inherent selection bias towards the null hypothesis among donor characteristics in the NRP cohort. Importantly, NRP is a technique that is exclusively used in the DCD setting, where it is well known that the incidence of DGF can be as high as three-fold compared to DBD grafts [84]. Studies involving DCD grafts maintained by NRP have shown that DGF rates remain elevated, which may be attributed to the higher severity of IRI that ensues and is more of a reflection of the DCD status rather than effects of the perfusion technique [13,61]. A meta-analysis comparing outcomes of grafts procured from DCD compared to DBD found that there were higher rates of DGF (RR 2.02; 95%CI 1.88, 2.16), PNF (RR 1.43; 95%CI 1.26, 1.63), 12-month graft loss (RR 1.13; 95%CI 1.08, 1.19), and 12-month patient mortality (RR 1.10; 95%CI 1.01, 1.21) [85]. Taken together, it is reasonable to conclude that organs procured by NRP can achieve non-inferior outcomes compared to SCS. The potential use of NRP is also intriguing in that it also optimizes the graft function of all other organs that are being procured while also minimizing IRI severity [86]. Presently, there are no randomized controlled trials on NRP, and future research should be encouraged in order to further delineate its limitations and benefits [87].

### 4.2. Future Directions: Perfusion Therapy, Antioxidants, and Technological Advances

#### 4.2.1. Incorporating Therapeutic Agents in Perfusate Solution 

Along with advances in dynamic perfusion devices and optimization of temperature regulation to mitigate the pathophysiological pathway of IRI, enhancing the effect of the preservation solution itself will also have an important role. Currently available preservatives were developed with the following considerations: (1) osmotic agents to prevent interstitial and intracellular edema, (2) buffering agents to maintain pH, (3) substrates for ATP production following reperfusion, and (4) antioxidant reagents [88]. As the use of perfusion technology becomes more widespread, future research should look towards innovative ways to reduce oxidative stress and IRI through advances in perfusate solution. Perfusion devices provide an opportunity to deliver therapeutic agents directly to the isolated organ and avoid limitations associated with systemic treatment. The metabolic and physiological conditions maintained during dynamic perfusion provides for the ideal environment for drug delivery without the use of viral conduits or other delivery methods that may incite an inflammatory response among immunosuppressed transplant recipients [89]. Thompson et al. demonstrated that MP can be used to deliver oligonucleotides to target microRNA function that minimizes IRI [89]. Therapeutic agents such as antioxidants [90], vasodilators [91,92], and thrombolytics [93,94] have also been found to show promise in improving graft outcomes, though more data are required in its application for larger, more heterogenous populations [95]. Advances in the development of extracellular hemoglobin, such as the M101 (HEMO_2_Life^®^, Hemarina, Morlaix, France) medical device, that feature high oxygen-carrying capabilities to directly combat organ ischemia can also be incorporated into dynamic perfusion techniques [96,97]. The OXYgen carrier for Organ Preservation (OxyOp) multicenter trial reported a lower risk of DGF by 66% and better short-term renal function among allografts treated by machine-perfused M101 [96]. The OxyOp2 follow-up trial is an expansive, multicenter effort that will aim to definitively test and quantify the efficacy of M101 for organ preservation in a large, heterogenous population [98].

Antioxidants also represent an important class of small molecules that can be integrated into dynamic perfusion techniques as they function to scavenge free radicals and ROS. Quercetin, hydrogen sulfide (HS), Tempol, Mito-TEMPO XJB-5-131, and MitoQ are just a few antioxidants that have been studied in pre-clinical models of IRI or transplantation. Quercetin is a polyphenol molecule belonging to the flavonoid family, which has been shown to alleviate oxidative stress and prevent apoptosis [99]. It has also been studied in a porcine autologous renal transplant model with simultaneous hypothermic oxygenated MP and shown to improve renal function through mediation of oxidative stress [100]. HS is a gaseous transmitter that has been shown to minimize oxidative stress through direct scavenging of ROS or increased production of glutathione [7]. Using an allogeneic rat kidney transplant model, Lobb et al. illustrated that HS supplementation in University of Wisconsin (UW) solution improved early allograft function and decreased necrosis and apoptosis of the renal allograft [101]. This group additionally used transcriptomics to show decreased expression of genes associated with renal injury, coagulation, and cellular stress responses. Similar results were observed in DCD porcine kidneys preserved by hypothermic UW solution supplemented with HS [102]. Although promising, a limitation of both quercetin and HS is that they have minimal mitochondrial penetrance. As such, molecules that are able to more effectively target mitochondria, like Tempol, Mito-TEMPO, XJB-5-131, and MitoQ, may have a greater impact on mitigating metabolic stress and mitochondrial ROS production [7]. Though they each belong to a different class of small molecule, they have been shown to decrease oxidative stress with improvement of renal function in small rodent kidney IRI models [99,103,104,105,106]. As these antioxidants continue to show promise in the pre-clinical setting, future efforts involving clinical trials are warranted.

Lastly, the use of mesenchymal stromal cell-derived extracellular vesicles (MSC-EVs) in dynamic perfusion techniques of kidney allografts also shows promise in mitigating immunogenicity [107,108,109]. Bone-marrow-derived mesenchymal stromal cells (MSCs) can secrete trophic factors and deliver extracellular vesicles (EVs) that repair tissue after injury and downregulate the alloimmune response [107,108,110,111]. A large RCT involving 156 living-donor kidney transplant patients showed that MSC infusion at kidney reperfusion resulted in lower rates of acute rejection, opportunistic infection, and better estimates of renal function at 1 year [112]. Moreover, pre-clinical models of organ transplantation have shown that MSCs can induce long-term graft tolerance in combination with immunosuppressive drugs [109]. However, it is also important to note that MSCs may also inadvertently promote a proinflammatory response and worsen allograft survival [109]. This may be the reason why the benefits of MSC-EV within the current kidney transplant literature are mixed, despite promising results in cohorts with chronic and acute kidney disease [108,113]. Ongoing trials are necessary to elucidate the evidence regarding the long-term safety of MSC therapy and determine the optimal cell source and infusion protocols to optimize outcomes in kidney transplant recipients [109].

#### 4.2.2. Alternative Clinical End Points for Future Dynamic Perfusion Research

Along with advances in the concoction of therapeutic perfusate, different outcome measurements can also evolve in the setting of increased data collection [114]. Assessment of noninvasive biomarkers and other inflammatory profiles may assist in anticipating the clinical trajectory of a graft prior to the occurrence of conventional endpoints (e.g., graft survival, patient survival). Interestingly, we found no difference in serum creatinine or eGFR at 3-, 6-, and 12-month time points between comparison groups. This may be due to a statistical type 2 error and lack of the required sample size to reject the null hypothesis, since these biomarkers were rarely the focus of study endpoints. Other emerging biomarkers that have been proposed for early identification of DGF include “omics” data: proteomics, transcriptomics, and metabolomics [115]. Along with noninvasive biomarkers identified in the post-transplant setting, MP can also be used as a preoperative tool to select and test marginal grafts under more morphological conditions by evaluating the biomarkers present in its effluent prior to transplantation. Tozzi et al. found that inflammatory markers such as TNF-alpha, IL-1β, IL-2, and sICAM-1 in the liquid effluent after procurement were reduced after MP compared to SCS [9]. Additionally, although there currently is a gap in the literature of objective perfusion parameters to predict post-transplant graft function, other studies have examined the utility of measuring the flow rate and vascular resistance during perfusion. For example, in a single-center RCT, Meister et al. found lower decreases in renal vascular resistance were associated with higher rates of DGF [51]. Future studies that investigate the utility of noninvasive biomarkers and perfusion parameters can provide further insight towards the different ways of leveraging the full capabilities of MP and NRP.

The results of our expansive systematic review and meta-analysis should be interpreted within the context of its limitations. First, because only peer-reviewed articles were included, our study may be susceptible to publication bias, as it does not include unpublished reports, meeting abstracts, or preprint letters. Next, there may be a fair amount of selection bias as some countries limit the amount of DCD grafts that can be accepted for transplantation. Though DCD grafts have acceptable outcomes, studies have shown that they may have slightly higher rates of PNF, DGF, and graft loss compared to DBD grafts [85]. While our analysis included studies involving both DCD and DBD grafts, we aimed to assess the outcomes of organ preservation techniques irrespective of the way kidney grafts were procured. Lastly, we did not account for each different type of post-transplant immunosuppressive regimen. However, this allows our results to be even more generalizable across different post-transplant practices.

## 5. Conclusions

This systematic review and meta-analysis highlight the benefits of MP and NRP compared to traditional preservation techniques. MP can overcome the risks of lower quality allografts and improve short- and long-term function compared to SCS. NRP reduces the rates of acute rejection compared to ISP, although future work is still required to elucidate its benefits, especially in the setting of randomized controlled trials. In all, MP and NRP are important organ preservation techniques to mitigate IRI and expand the donor pool to satisfy the increasing demand of kidney transplantation. The clinical benefits of MP and NRP should encourage its use as the standard of care for kidney allograft preservation.

## Figures and Tables

**Figure 1 antioxidants-13-00642-f001:**
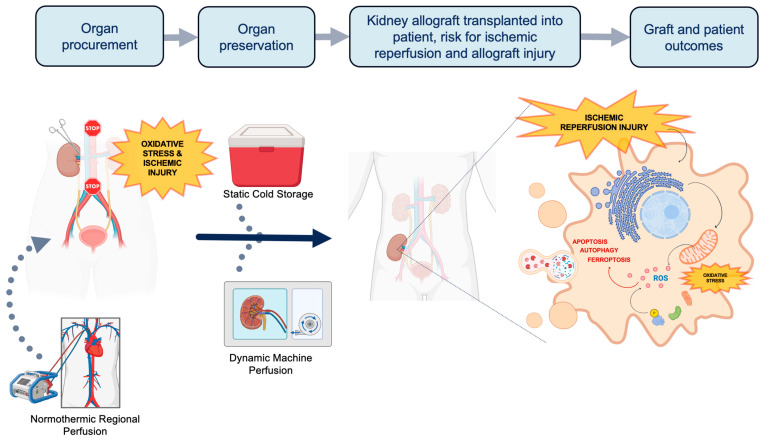
The process of organ procurement and the potential of implementing normothermic regional perfusion (NRP) and machine perfusion (MP) techniques. Traditionally, organs procured for transplantation are preserved in static cold storage (SCS). Transplanted organs suffer two insults of ischemic injury and oxidative stress: once during procurement and another during implantation. At the time of implantation, ischemic reperfusion injury (IRI) triggers the activation of proteinases, increases the permeability of the mitochondrial membrane, and generates reactive oxygen species. This process further exacerbates oxidative stress, apoptosis, ferroptosis and autophagy. NRP and MP are dynamic perfusion techniques that can reduce oxidative stress and IRI through unique mechanisms. NRP can be applied prior to and during organ procurement, while HMP is used as a mode of organ preservation. Both techniques also hold potential in therapeutic drug delivery and expanding the donor pool (Figure created with www.biorender.com (accessed on 5 April 2024)).

**Figure 2 antioxidants-13-00642-f002:**
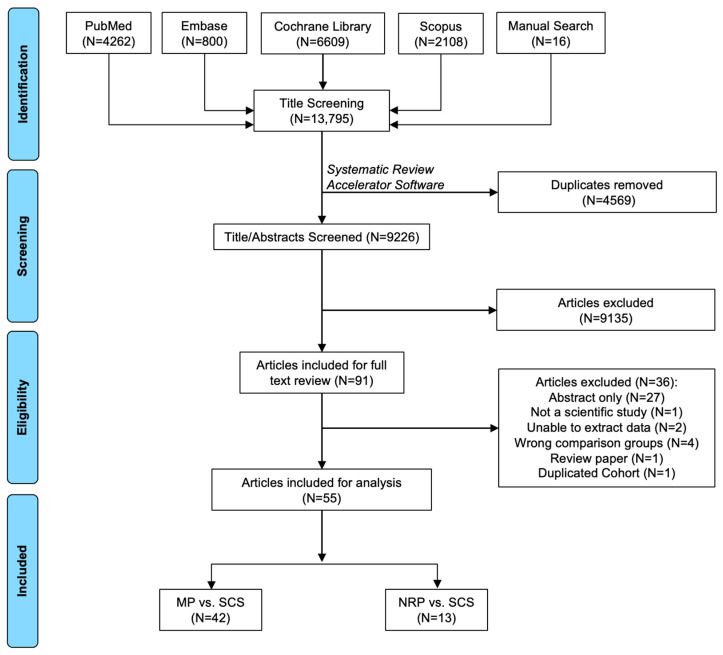
Search results and application of eligibility criteria.

**Figure 3 antioxidants-13-00642-f003:**
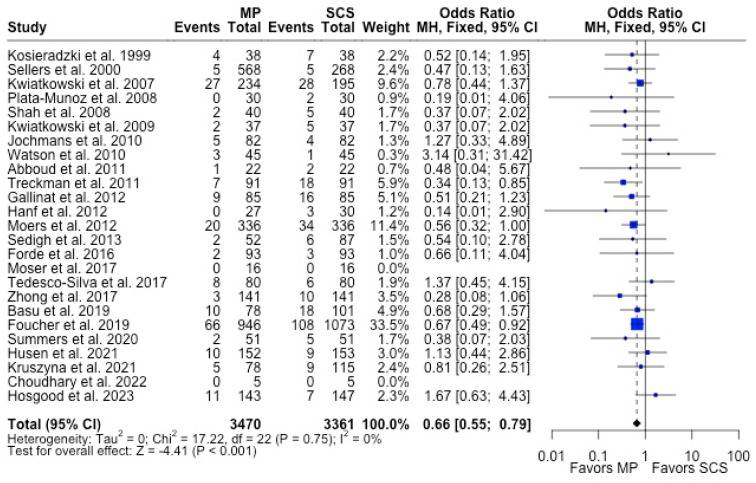
Forest plot for rates of 12 months graft survival between MP and SCS [15,17,24,25,26,27,29,30,31,32,33,34,35,36,38,42,43,44,46,49,50,52,53,54,57].

**Figure 4 antioxidants-13-00642-f004:**
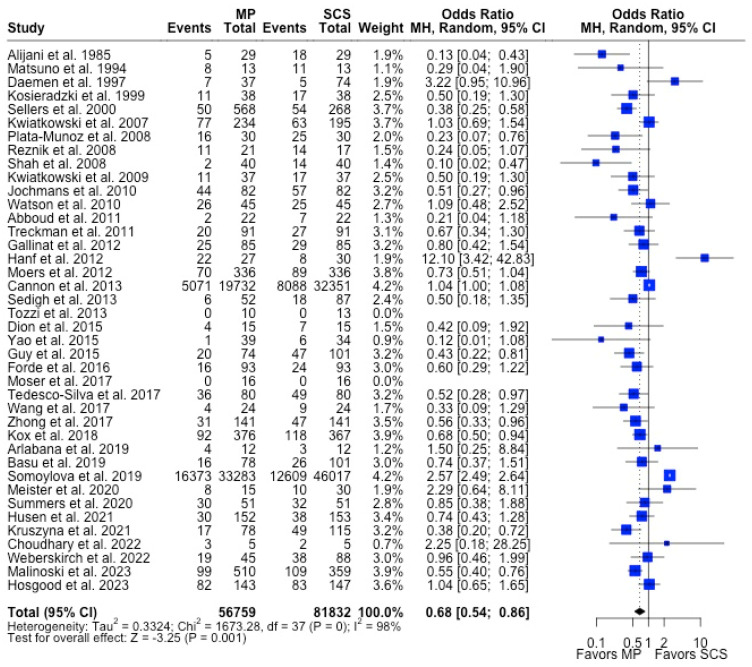
Forest plot for rates of DGF between MP and SCS [9,15,17,20,22,23,24,25,26,27,28,29,30,31,32,33,34,35,36,37,38,39,40,41,42,43,44,45,46,47,48,49,51,52,53,54,55,56,57,71].

**Figure 5 antioxidants-13-00642-f005:**
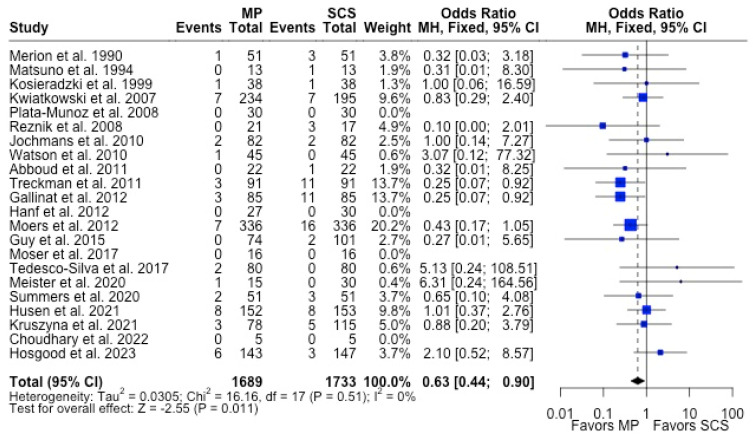
Forest plot for rates of PNF between MP and SCS [15,17,21,22,24,26,27,28,31,32,33,34,35,36,40,43,44,51,52,53,54,57].

**Figure 6 antioxidants-13-00642-f006:**
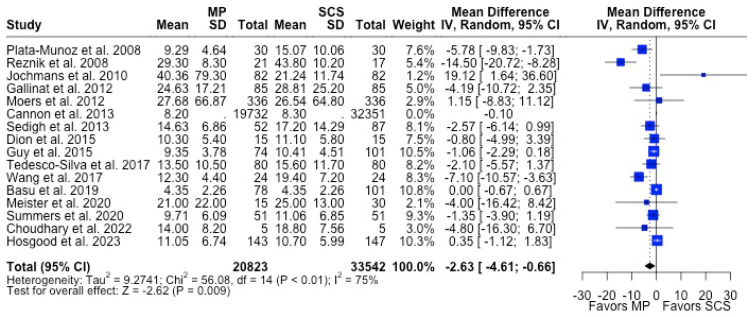
Forest plot for mean of length of hospital stay between MP and SCS [15,27,28,31,34,36,37,38,39,40,44,45,49,51,52,57].

**Figure 7 antioxidants-13-00642-f007:**
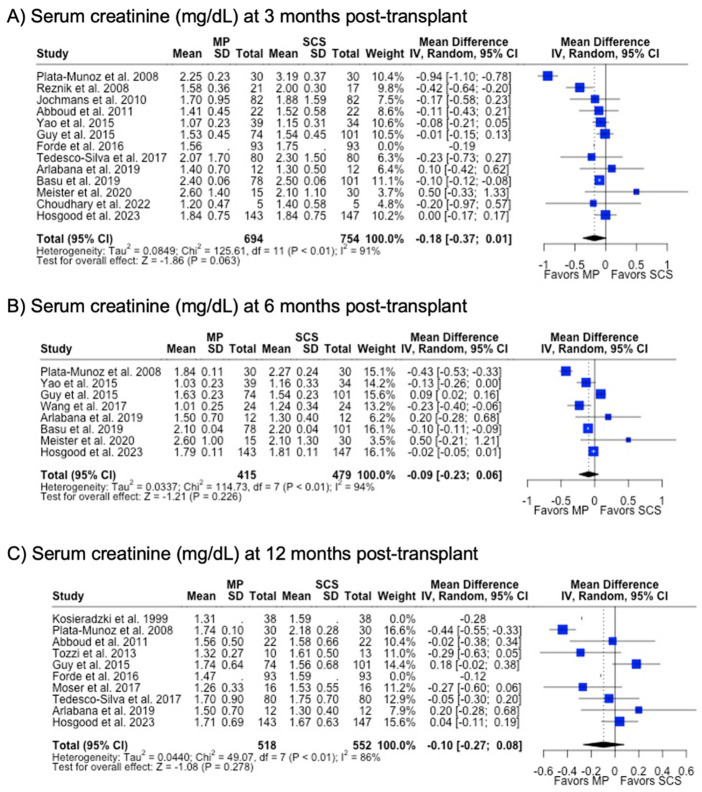
Forest plot for post-transplant serum creatinine (mg/dL) between MP and SCS [9,15,17,24,27,28,31,40,41,42,43,44,45,48,49,51,57].

**Figure 8 antioxidants-13-00642-f008:**
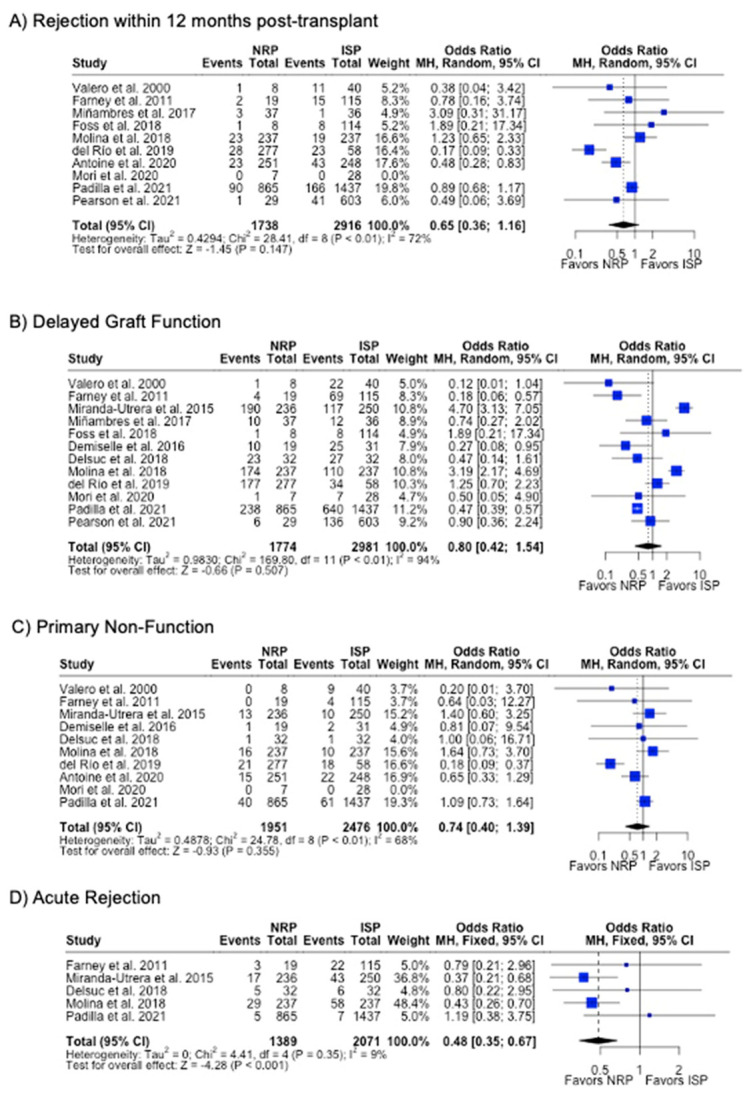
Forest plot for rates (**A**) acute rejection, (**B**) delayed graft function, (**C**) primary non-function, (**C**) acute rejection, and (**D**) graft failure within 12 months post-transplant between NRP and ISP [58,59,60,61,62,63,64,65,66,67,68,69,70].

**Figure 9 antioxidants-13-00642-f009:**
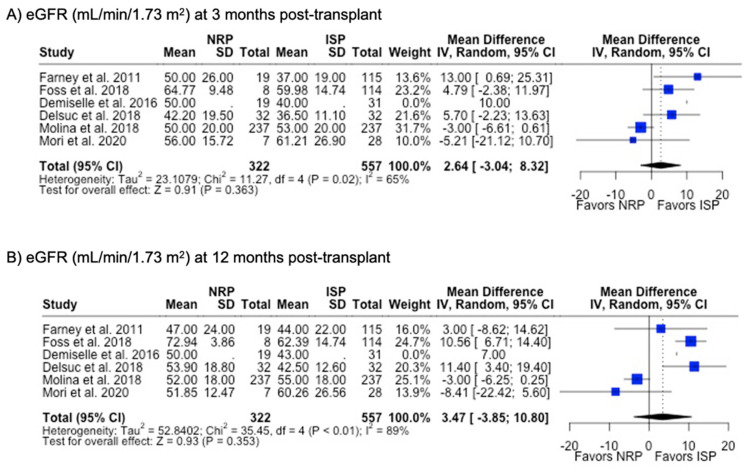
Forest plot for post-transplant eGFR (mL/min/1.73 m^2^) at (**A**) 3 months and (**B**) 12 months between NRP and ISP [59,62,63,64,66,67].

**Table 1 antioxidants-13-00642-t001:** Pooled descriptive statistics of clinicodemographic characteristics and outcomes of patients involved in MP vs. SCS studies.

Characteristics	Total Studies (N = 42)	MP (N = 57,756)	SCS (N = 82,956)	*p*-Value
Age (years), mean (95%CI)				
Donor	32	50.0 (46.3, 53.8)	49.8 (45.7, 53.8)	0.702
Recipient	37	50.6 (47.7, 53.5)	50.8 (48.0, 53.7)	0.776
Cold Ischemic Time (h), mean (95%CI)	31	15.4 (12.8, 18.0)	13.4 (11.0, 15.7)	0.005
Warm Ischemic Time (min), mean (95%CI)	11	33.9 (25.4, 42.4)	30.8 (23.1, 38.6)	0.016
Donor Serum Creatinine (mg/dL), mean (95%CI)	17	1.16 (1.15, 1.18)	1.09 (1.08, 1.10)	0.030
Outcomes				
Length of Stay (days), mean (95%CI)	15	15.5 (10.7, 20.4)	18.4 (13.0, 23.7)	0.009
DGF, %	40	27.7% (21.9%, 34.3%)	36.4% (30.3%, 43.1%)	0.001
Duration of DGF (days), mean (95%CI)	11	6.4 (6.1, 6.8)	9.9 (9.5, 10.2)	0.452
Number of Dialysis, mean (95%CI)	4	3.7 (1.8, 5.5)	2.5 (2.2, 2.8)	0.938
PNF, %	18	2.8% (2.1%, 3.7%)	4.4% (3.6%, 5.5%)	0.011
Acute Rejection, %	22	12.3% (8.5%, 17.5%)	14.9% (10.7%, 20.4%)	0.270
Graft Failure, %				
12 months	23	5.3% (3.9%, 7.2%)	8.3% (6.4%, 10.8%)	<0.001
3 Years	6	10.6% (5.2%, 20.5%)	19.3% (12.6%, 28.5%)	0.128
Serum Creatinine (mg/dL), mean (95%CI)				
3 months	12	1.74 (1.44, 2.03)	1.90 (1.52, 2.27)	0.063
6 months	8	1.67 (1.24, 2.10)	1.70 (1.32, 2.08)	0.226
12 months	8	1.58 (1.41, 1.74)	1.66 (1.44, 1.88)	0.278
eGFR (mL/min/1.73 m^2^), mean (95%CI)				
3 months	10	47.4 (38.7, 56.1)	47.7 (37.5, 57.9)	0.985
6 months	6	52.6 (25.9, 79.2)	49.8 (30.7, 68.9)	0.318
12 months	8	48.6 (42.0, 55.2)	45.8 (42.5, 49.0)	0.161
Mortality, %				
12 months	18	4.1% (3.3%, 5.0%)	3.6% (2.5%, 5.1%)	0.415

MP = machine perfusion, SCS = static cold storage, 95%CI = 95% confidence interval, ECD = expanded criteria donors, DGF = delayed graft function, PNF = primary nonfunction, eGFR = estimated glomerular filtration rate.

**Table 2 antioxidants-13-00642-t002:** Pooled descriptive statistics of clinicodemographic characteristics and outcomes of patients involved in NRP vs. ISP studies.

Characteristics	Total Studies (N = 13)	NRP (N = 2025)	ISP (N = 3229)	*p*-Value
Age, mean (95%CI)				
Donor	9	48.1 (42.9, 53.2)	47.8 (41.6, 54.0)	0.903
Recipient	8	50.1 (46.2, 53.9)	49.4 (45.2, 53.7)	0.042
Cold Ischemic Time (h), mean (95%CI)	8	12.7 (10.0, 15.4)	17.2 (13.4, 21.1)	0.001
Warm Ischemic Time (min), mean (95%CI)	2	22.5 (22.1, 23.0)	24.4 (24.0, 24.8)	<0.001
Donor Serum Creatinine (mg/dL), mean (95%CI)	5	1.34 (1.09, 1.58)	1.29 (0.85, 1.73)	0.639
Outcomes				
Length of Stay (days), mean (95%CI)	3	6.88 (6.5, 7.3)	11.9 (11.1, 12.7)	0.347
DGF, %	12	40.1% (25.5%, 57.9%)	45.4% (31.1%, 60.5%)	0.507
PNF, %	10	5.48% (4.6%, 6.6%)	6.4% (3.5%, 11.4%)	0.355
Acute Rejection, %	5	6.4% (2.1%, 17.7%)	10.0% (2.7%, 30.5%)	<0.001
Graft Failure, %				
12 months	10	9.9% (8.6%, 11.4%)	11.1% (6.6%, 17.9%)	0.147
3 Years	4	15.2% (11.6%, 19.8%)	18.4% (5.5%, 46.6%)	0.546
Serum Creatinine (mg/dL), mean (95%CI)				
12 months	3	1.43 (1.17, 1.69)	1.58 (1.00, 2.16)	0.217
eGFR (mL/min/1.73 m^2^), mean (95%CI)				
3 months	5	49.3 (47.0, 51.6)	46.4 (27.3, 65.5)	0.363
12 months	5	52.0 (49.9, 54.1)	49.9 (36.6, 63.1)	0.353
Mortality, %				
12 months	4	1.3% (0.16%, 9.6%)	3.2% (0.6%, 14.5%)	0.106
3 Years	3	1.4% (0.5%, 3.6%)	2.0% (0.3%, 13.7%)	0.745

NRP = normothermic regional perfusion, ISP = in situ cold preservation, 95%CI = 95% confidence interval, ECD = expanded criteria donors, DGF = delayed graft function, PNF = primary nonfunction, eGFR = estimated glomerular filtration rate.

## Data Availability

Not applicable.

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
