# Peer review of "Perfusion Techniques in Kidney Allograft Preservation to Reduce Ischemic Reperfusion Injury: A Systematic Review and Meta-Analysis"

_antioxidants, 2024, doi:10.3390/antiox13060642_

Round 1
Reviewer 1 Report
Comments and Suggestions for Authors
I read with great interest the paper by Bima J. Hasjim et al. who conducted a systematic review and meta-analysis to evaluate the benefits of MP and NFP compared to conventional preservation techniques in DCD or ECD kidneys.The paper is well written and easy to understand, and the figures are clear. In my opinion, the discussion should be expanded as the therapeutic effects of MSC and EV in preclinical models of DCD donor perfusion are not mentioned in section 4.2.1 "Incorporation of therapeutic agents into the perfusate solution"
Comments on the Quality of English LanguageMinor editing of English language required
Author Response
We thank the editor and reviewers for thoughtful comments and positive feedback on our manuscript, which have helped us to improve the manuscript. We have revised the manuscript based on the reviewers’ comments and suggestions. Our point-to-point responses to individual comments are detailed below. We highlighted the changes we made in the resubmitted vision.
Comments from the editors and reviewers:
Review 1
1. I read with great interest the paper by Bima J. Hasjim et al. who conducted a systematic review and meta-analysis to evaluate the benefits of MP and NFP compared to conventional preservation techniques in DCD or ECD kidneys. The paper is well written and easy to understand, and the figures are clear. In my opinion, the discussion should be expanded as the therapeutic effects of MSC and EV in preclinical models of DCD donor perfusion are not mentioned in section 4.2.1 "Incorporation of therapeutic agents into the perfusate solution"
Thank you for your comments. We share the reviewer’s enthusiasm and agree that including a discussion on the therapeutic effects of mesenchymal stromal cells and extracellular vesicles will enhance the quality of the manuscript. We have included the following into section 4.2.1 Incorporating therapeutic agents in perfusate solution:
Lastly, the use of mesenchymal stromal cell-derived extracellular vesicles (MSC-EVs) in dynamic perfusion techniques of kidney allografts also show promise in mitigating immunogenicity [1–3]. Bone marrow-derived mesenchymal stromal cells (MSCs) can secrete trophic factors, and deliver extracellular vesicles (EVs) that repair tissue after injury and downregulate the alloimmune response [1,2,4,5]. A large RCT involving 156 living-donor kidney transplant patients showed that MSC infusion at kidney reperfusion resulted in lower rates of acute rejection, opportunistic infection, and better estimates of renal function at 1 year [6]. Moreover, pre-clinical models of organ transplantation have shown that MSCs can induce long-term graft tolerance in combination with immunosuppressive drugs [3]. However, it is also important to note that MSCs may also inadvertently promote a proinflammatory response and worsen allograft survival [3]. This may be the reason why the benefits of MSC-EV within the current kidney transplant literature are mixed, despite promising results in cohorts with chronic and acute kidney disease [2,7]. Ongoing trials are necessary to elucidate the evidence regarding long-term safety of MSC therapy and determine the optimal cell source and infusion protocols to optimize outcomes in kidney transplant recipients [3].
Review 2
1. I considered the manuscript entitled “ Perfusion Techniques in Kidney Allograft Preservation to Reduce Ischemic Reperfusion Injury and Oxidative Stress: A Systematic Review and Meta-Analysis” by Bima J. Hasjim, et al, which is intended to be published in Antioxidant journal.
I enjoyed the meta-analysis as it is well focused, well performed, and well described. The findings are those that can be found in the literature at this time. The study gives a measured evaluation of the pooled data in huge populations of transplanted patients suffering the classic or the new methods of preservation.
No concerns for its publication, without any modification of the manuscript
Thank you for this feedback. We hope that our study can continue to advocate for the use of dynamic perfusion techniques to mitigate ischemic reperfusion injury and oxidative stress to expand the donor pool for kidney transplantation.
Review 3
1. The article is interesting and very carefully prepared.
However, I doubt whether it suits this journal and this Special Issue.
Of course, it is based on Ischemia-reperfusion Injury - but I did not find any aspects of oxidative stress in it.
Also, looking at the keywords, it is clear that there are no aspects of oxidative stress.
The authors in the Introduction mention that prolonged ischemic conditions lead to oxidative stress, and that's it. Then, in 4.2.1. "antioxidants" appear. Generally, I think this is a well-prepared article, but the aspects of oxidative stress are not crucial here.
I think the authors should better justify why this article is appropriate for this journal.
Also, the goal of comparing the benefits of MP and NRP vs. traditional preservation techniques does not explain why the article should be published in Antioxidants.
Thank you for your comments.
This is an invited manuscript for this special issue: “New Strategies in Preventing Inflammatory and/or Oxidative-Stress-Induced Damages in Ischemia–Reperfusion Injury”.
This is this special issue information from the web site of “Antioxidants”
The purpose of this Special Issue of Antioxidants is to show the most recent advances in the application of therapies aimed at minimizing, reducing or even preventing IRI damage in different organs or systems, regardless of its causes. These therapies may involve both well-known natural and synthetic antioxidants, as well as new options based on the administration of stem cells or its derivative products (secretome) as therapeutic strategies. This Special Issue calls for original research papers and in-depth reviews that address i) a better understanding of the underlying molecular processes involved in IRI; ii), the progress and current status of the therapeutic management of IRI; and iii) aspects related to the preservation of the organ function, due to their special clinical interest.
We removed oxidative stress from the title.
We also found related studies published by “Antioxidants”:
https://www.mdpi.com/2076-3921/12/1/31
https://www.mdpi.com/2076-3921/10/8/1263
Personally, I was recently invited by Antioxidants as a guest editor for special Issue "Anti-Oxidative Therapy in Organ Transplantation.
We believe Antioxidants journal is interested in this topic and seeks articles in this field. We think our draft fits this specific issue well.
We also modified sections and Figure below
Section 1.1 in the Introduction (half page) is dedicated towards framing the clinical implications of organ preservation techniques within the context of ischemic reperfusion injury and oxidative stress
- Figure 1 is dedicated to showing the molecular effects of how different organ preservation techniques can mitigate ischemic reperfusion injury or oxidative stress
- Section 4.2.1 in the Discussion (1.5 pages) is dedicated towards future directions in dynamic perfusion therapies. Namely, in the delivery of therapeutic drugs such as antioxidants, micro RNA, mesenchymal stromal cells and extracellular vesicles.
2. Technical errors - unnecessary $ signs appear - "Although the initial investment in MP devices may be costly in the perioperative period ($1,182 MP vs $234 SCS), estimates of costs at 12-month follow-up showed that MP was ultimately $3,627 cheaper than SCS per transplant ($8,668 vs. $11,394) [77]."
Thank you for bringing this matter to our attention. The “$” signs were used in accordance with the guidelines for the journal. However, since it seems like its use may be distracting, we have removed any unnecessary “$” signs in this next draft. Below is the revised excerpt and we are open to future adjustments that align with the final format of the journal.
Despite the costly investment of MP devices in the perioperative period ($1,182 MP vs 234 SCS), estimates of costs at 12-month follow-up showed that MP was ultimately $3,627 cheaper than SCS per transplant ($8,668 vs. 11,394) [8]. Gomez et al. estimated as much as $3,369 in savings for each DGF or PNF avoided by MP [9]. The main contributors to the disparity in costs between MP and SCS were from post-transplant dialysis requirements ($4,390 MP vs. 7,581 SCS) and hospital readmissions ($2,892 MP vs. 3,174 SCS) [8].
3. The authors use the term creatine instead of creatinine, which should be corrected.
Thank you for the opportunity to correct this mistake. All instances where creatine was used, instead of creatinine, have been fixed.
4. A list of abbreviations would be helpful to make the article more accessible for the reader to understand.
Thank you for this feedback. We have included the following list of abbreviations to the manuscript:
95%CI: 95% Confidence Interval
ATP: adenosine triphosphate
cDCD: controlled donors after circulatory death
CIT: cold ischemia time
CKD: chronic kidney disease
DBD: donation after brain death
DCD: donation after circulatory death
DGF: delayed graft function
ECD: extended criteria donors
eGFR: estimated glomerular filtration rate
EV: extracellular vesicle
HMP: hypothermic machine perfusion
HS: hydrogen sulfide
IL: interleukin
IRI: ischemic reperfusion injury
ISP: in-situ cold preservation
MP: machine perfusion
MSC: mesenchymal stromal cells
NMP: normothermic machine perfusion
NRP: normothermic regional perfusion
OR: odds ratio
OxyOp: Oxygen carrier for Organ Preservation
PNF: primary non-function
PRIMSA-P: Preferred Reporting Items for Systematic Reviews and Meta-analyses Protocols
RCT: randomized controlled trial
RNA: ribonucleic acid
ROS: reactive oxygen species
RR: risk ratio
SCS: static cold storage
siCAM: soluble intercellular adhesion molecule-1
TNF: tumor necrosis factor
uDCD: uncontrolled donors after circulatory death
UW: University of Wisconsin
WIT: warm ischemia time
Reviewer 2 Report
Comments and Suggestions for Authors
I considered the manuscript entitled “ Perfusion Techniques in Kidney Allograft Preservation to Reduce Ischemic Reperfusion Injury and Oxidative Stress: A Systematic Review and Meta-Analysis” by Bima J. Hasjim, et al, which is intended to be published in Antioxidant journal.
I enjoyed the meta-analysis as it is well focused, well performed, and well described. The findings are those that can be found in the literature at this time. The study gives a measured evaluation of the pooled data in huge populations of transplanted patients suffering the classic or the new methods of preservation.
No concerns for its publication, without any modification of the manuscript
Author Response

(The authors gave the same response as above.)

Reviewer 3 Report
Comments and Suggestions for Authors
The article is interesting and very carefully prepared.
However, I doubt whether it suits this journal and this Special Issue.
Of course, it is based on Ischemia-reperfusion Injury - but I did not find any aspects of oxidative stress in it.
Also, looking at the keywords, it is clear that there are no aspects of oxidative stress.
The authors in the Introduction mention that prolonged ischemic conditions lead to oxidative stress, and that's it. Then, in 4.2.1. "antioxidants" appear. Generally, I think this is a well-prepared article, but the aspects of oxidative stress are not crucial here.
I think the authors should better justify why this article is appropriate for this journal.
Also, the goal of comparing the benefits of MP and NRP vs. traditional preservation techniques does not explain why the article should be published in Antioxidants.
Technical errors - unnecessary $ signs appear - "Although the initial investment in MP devices may be costly in the perioperative period ($1,182 MP vs $234 SCS), estimates of costs at 12-month follow-up showed that MP was ultimately $3,627 cheaper than SCS per transplant ($8,668 vs. $11,394) [77]."
The authors use the term creatine instead of creatinine, which should be corrected.
A list of abbreviations would be helpful to make the article more accessible for the reader to understand.
Author Response

(The authors gave the same response as above.)

Round 2
Reviewer 3 Report
Comments and Suggestions for Authors
The article has been very well corrected. All my comments have been considered, and I am fully satisfied with the changes. I think that the article in the current version can be accepted. I recommend to consider it for publication